# Protective Effects of Traditional Polyherbs on Cisplatin-Induced Acute Kidney Injury Cell Model by Inhibiting Oxidative Stress and MAPK Signaling Pathway

**DOI:** 10.3390/molecules25235641

**Published:** 2020-11-30

**Authors:** VinayKumar Dachuri, Phil Hyun Song, Young Woo Kim, Sae-Kwang Ku, Chang-Hyun Song

**Affiliations:** 1Department of Anatomy and Histology, College of Korean Medicine, Daegu Haany University, Gyeongsan 38610, Korea; dachurivinay@dhu.ac.kr; 2Research Center for Herbal Convergence on Liver Disease, College of Korean Medicine, Daegu Haany University, Gyeongsan 38610, Korea; 3Department of Urology, College of Medicine, Yeungnam University, Daegu 42415, Korea; sph04@hanmail.net; 4School of Korean Medicine, Dongguk University, Gyeongju 38066, Korea; ywk@dongguk.ac.kr

**Keywords:** acute kidney injury, acute renal failure, renal cell injury, polyherb, cisplatin, nephroprotective, antioxidant, antiapoptosis, MAPK

## Abstract

Acute kidney injury (AKI) is a disease caused by sudden renal dysfunction, which is an important risk factor for chronic renal failure. However, there is no effective treatment for renal impairment. Although some traditional polyherbs are commercially available for renal diseases, their effectiveness has not been reported. Therefore, we examined the nephroprotective effects of polyherbs and their relevant mechanisms in a cisplatin-induced cell injury model. Rat NRK-52E and human HK-2 subjected to cisplatin-induced AKI were treated with four polyherbs, Injinhotang (IJ), Ucha-Shinki-Hwan (US), Yukmijihwang-tang (YJ), and Urofen^TM^ (Uro) similar with Yondansagan-tang, for three days. All polyherbs showed strong free radical scavenging activities, and the treatments prevented cisplatin-induced cell death in both models, especially at 1.2 mg/mL. The protective effects involved antioxidant effects by reducing reactive oxygen species and increasing the activities of superoxide dismutase and catalase. The polyherbs also reduced the number of annexin V-positive apoptotic cells and the expression of cleaved caspase-3, along with inhibited expression of mitogen-activated protein kinase-related proteins. These findings provide evidence for promoting the development of herbal formulas as an alternative therapy for treating AKI.

## 1. Introduction

Acute kidney injury (AKI), also known as acute renal failure, is a disease caused by sudden abruption in renal function with severe tubular damage, which is considered a global health problem that accounts for 9.5% of in-hospital mortality [1]. Its etiology includes infections, sepsis, failure of renal cell repair, and nephrotoxic drug insults. Cisplatin is one of the most effective chemotherapeutic agents for treating various cancers of the ovary, head and neck, lung, breast, and bladder; however, its application is limited by the development of nephrotoxicity [2]. Indeed, AKI occurs in 20–30% of patients treated with cisplatin [3]. The main treatments for AKI focus on symptomatic therapies including saline hydration and diuresis, and discontinuation of cisplatin and other specific drugs until renal function is recovered. Considering that AKI increases the risk of chronic or end-stage renal disease with an exceptionally high mortality rate, the socioeconomic importance is increasing [4]. End-stage renal disease can only be treated with dialysis and kidney replacement. Therefore, therapeutic strategies are urgently needed to reduce the risk of developing cisplatin-induced nephrotoxicity.

The pathological progress of AKI involves oxidative stress, apoptosis, and inflammation [5]. In particular, oxidative stress is closely associated with cisplatin-induced AKI; reactive oxygen species (ROS) overwhelming the endogenous antioxidant defense system induce lipid peroxidation and cell damage, resulting in a reduction in the glomerular filtration rate [6]. The overproduced ROS serves as intracellular signaling molecules to activate nuclear factor (NF)-κB [7] and mitogen-activated protein kinase (MAPK) signaling [8]. Indeed, cisplatin-induced renal tubular damage involves the activation of signaling proteins such as p38, extracellular signal-regulated kinase (ERK), and c-Jun N-terminal kinase (JNK) [9]. There have been many efforts to develop pharmacological agents for treating renal impairment, with growing interest in the beneficial effects of various antioxidants, such as vitamins C (ascorbic acid) and E, curcumin, selenium, and bixin [10]. Among these, ascorbic acid is a powerful antioxidant and free radical scavenger, which ameliorates renal failure and tubular cell damage in the cisplatin-induced AKI model [11,12]. Furthermore, several dietary phytochemicals containing polyphenols and flavonoids also appear to be efficient in AKI animal models [13]. However, most antioxidant agents have shown inconsistencies between preclinical and clinical studies despite their positive effects in the AKI model [10].

Conversely, there are traditional polyherb formulae used for treating renal deficiency since ancient times in East Asia, including Korea, China, and Japan. Some traditional polyherbs have been approved as pharmaceuticals and they are commercially available, based on long-term clinical records rather than scientific evidence; however, their effectiveness in different renal diseases needs to be clarified. Furthermore, their use should be carefully considered, because some herbal components (i.e., aristolochic acid, croton, podophyllotoxin, and other plant alkaloids) can induce severe nephrotoxicity [14]. There have been recent reports supporting the nephroprotective effects of *Boerhaavia diffusa, Rheum emodi, Nelumbo nucifera,* and *Crataeva nurvala*, as well as active compounds including curcumin and red ginseng extracts [15,16,17]. However, because most traditional herbal medicines are used as combination formulae consisting of various herbs for synergic effects, extensive studies are needed to prove their effectiveness for clinical use. Although mechanistic studies on individual herbs are difficult, recent systems pharmacology can provide an understanding of the nature of traditional medicines and their mechanisms [18,19].

We have attempted to screen the nephroprotective effects of polyherbs used for renal diseases in traditional Korean medicine [20], and from these, we selected four polyherbs showing strong antioxidant activities for this study. These polyherbs included the traditional medicines Injinhotang (IJ), Ucha-Shinki-Hwan (US; *Gochajinkigan* in Japanese), Yukmijihwang-tang (YJ; *Liuweidihuang-tang* in Chinese; *Rokumijio-to* in Japanese), and Urofen^TM^ (Uro) comprising ingredients similar to Yondansagan-tang (*Longdanxiegan-tang* in Chinese; *Ryutanshakan-to* in Japanese). The nephroprotective effects and relevant mechanisms were examined in a cisplatin-induced renal cell injury model.

## 2. Results

### 2.1. Free Radical Scavenging Activities of Traditional Polyherbs

In the results of the 1,1-diphenyl-2-picrylhydrazyl (DPPH) assay, one-way analysis of variance (ANOVA) showed significant differences among the groups (F = 121.7; *p* < 0.01, Figure 1). The post-hoc tests versus the vehicle control revealed significant decreases by 15.2%, 37.9%, 29.8%, and 28.3% in the IJ, Uro, US, and YJ groups, respectively (*p* < 0.01). The value was also decreased by 38.7% in the ascorbic acid (AA) group (*p* < 0.01).

### 2.2. Effects on Cell Viabilities in Cisplatin-Induced Nephrotoxicity

Compared to the normal control without cisplatin, the cell viabilities were 51.7% and 48.8% in the vehicle-treated (Veh) group of normal rat kidney (NRK) and human kidney (HK)-2 epithelial cells after cisplatin induction, respectively (*p* < 0.01), approximately reaching to the half-maximal inhibitory concentration (IC_50_) values (Figure 2). Two-way ANOVA showed significant main effects for the group in the NRK (F = 187.0; *p* < 0.01) and HK-2 (F = 101.1; *p* < 0.01), as well as for the dose in the NRK (F = 122.6; *p* < 0.01) and HK-2 (F = 119.2; *p* < 0.01). There were significant interactions between the group and dose in the NRK (F = 122.2; *p* < 0.01) and HK-2 (F = 250.7; *p* < 0.01), indicating dose-dependent differences among the groups. Compared to the Veh group in the AKI model, the cell viabilities significantly increased in treatments with IJ at 0.6 and 1.2 mg/mL, Uro at 0.3–2.4 mg/mL, US at 1.2 and 2.4 mg/mL, YJ at 0.3–2.4 mg/mL, and AA at 35–18 μg/mL in the NRK model (*p* < 0.01). The increases were found in all the treatments at 0.6 and 1.2 mg/mL in the HK-2 (*p* < 0.01). The cell viabilities were increased more with the polyherbs at 1.2 mg/mL and AA at 35 μg/mL compared to the other doses in both models, and the treatments were, thus, used at these doses for further experiments.

### 2.3. Effects on ROS Production and Antioxidant Activities

The cells stained with dichlorofluorescein diacetate (H2DCFDA) for ROS levels were evidently more in the cisplatin-treated Veh group of the NRK and HK-2; however, these seemed to be fewer in the treatments with polyherbs (Figure 3). Few stained cells were observed in the normal control of the NRK and HK-2 cells without cisplatin treatment, regardless of the groups. One-way ANOVA showed significant differences among the groups for ROS in the NRK (F = 39.4; *p* < 0.01) and HK-2 models (F = 53.1; *p* < 0.01). The levels of ROS in the NRK and HK-2 increased 2.5- and 3.1-fold in the cisplatin-treated Veh group, respectively, compared to that in the corresponding cisplatin non-treated controls (*p* < 0.01). However, the post-hoc tests versus the cisplatin-treated Veh group revealed significant decreases of 16.6%, 13.9%, 21.5%, 26.5%, and 21.9% in IJ, Uro, US, YJ, and AA groups in the NRK, and of 25.0%, 14.4%, 26.4%, 33.6%, and 34.1% in the HK-2, respectively (*p* < 0.01). In addition, significant differences were also observed among the groups for superoxide dismutase (SOD) activity in the NRK (F = 181.9; *p* < 0.01) and HK-2 (F = 130.3; *p* < 0.01), and for catalase activity in the NRK (F = 70.9; *p* < 0.01) and HK-2 (F = 102.3; *p* < 0.01). Compared to the cisplatin non-treated control, the cisplatin-treated Veh group showed significant decreases in the activities of SOD and catalase (*p* < 0.01, Figure 3): the SOD activity decreased by 39.5% and 41.2% and the catalase activity decreased by 39.9% and 38.6%, respectively in the NRK and HK-2 models. Compared to the cisplatin-treated Veh group, SOD activity significantly increased by 1.4 folds in the Uro, US, YJ, and AA groups, and 1.3 folds in the IJ in NRK, and by 1.4 folds in the US and 1.3 folds in the other groups in HK-2 (*p* < 0.01). Catalase activity increased by 1.4 folds in the IJ, US, and AA, and 1.3 folds in the Uro and YJ in NRK, and by 1.3 folds in the IJ and AA, and 1.2 folds in the other groups in HK-2 (*p* < 0.01). However, there were no significant differences among the groups in the cisplatin non-treated conditions.

### 2.4. Effects on Apoptotic Changes

Flow cytometric analysis showed that the number of annexin V-positive cells was low in the cisplatin non-treated NRK and HK-2, whereas they were higher in the cisplatin-treated Veh control of both cell models (Figure 4). However, the number of annexin V-positive cells tended to be lower in both cell models treated with the polyherbs. There were significant differences among the groups in NRK (F = 130.5; *p* < 0.01) and HK-2 (F = 197.9; *p* < 0.01). The post-hoc tests versus the cisplatin-treated Veh control revealed significant decreases by 37.6%, 53.9%, 46.5%, 64.3%, and 44.9% in the IJ, Uro, US, YJ, and AA groups, respectively, in the NRK, and by 24.6%, 47.6%, 45.4%, 54.0%, and 48.7% in the HK-2 (*p* < 0.01). However, there were no significant differences in the number of cells immunostained for annexin-V only.

### 2.5. Effects on Cell Proliferation

Cell proliferation was assessed in the cisplatin non-treated normal cells after treatment with the polyherbs for three days under serum-free conditions (Figure 5). Ki-67-positive cells showed no differences among the groups in the NRK; however, they were significantly different in the HK-2 (F = 16.6; *p* < 0.01). The post-hoc tests versus the normal Veh group showed significant increases by 1.2 folds in US and YJ (*p* < 0.01). Consistently, the 3-(4,5-Dimethylthiazol-2-yl)-2,5-diphenyltetrazolium bromide (MTT) assay for cell growth also showed that cells treated with US and YJ increased by 1.3 and 1.4 folds, respectively (*p* < 0.01).

### 2.6. Effects on Expression of Caspase-3 and MAPK Signaling Proteins

The expression levels of cleaved caspase-3 and the phosphorylated forms of p-38 (p38-α) and JNK (JNK2) increased upon cisplatin treatment (Figure 6). For cleaved caspase-3, the cisplatin-treated Veh group showed a significant increase by 3.0 folds compared to that in the cisplatin non-treated normal control (*p* < 0.01). There were significant differences among the groups in NRK (F = 97.8, *p* < 0.01) and HK-2 (F = 75.7, *p* < 0.01). The post-hoc tests versus the cisplatin-treated Veh group showed significant decreases by 49.4%, 27.3%, 45.1%, 30.0%, and 43.4% in the IJ, Uro, US, YJ, and AA groups, respectively, in the NRK, and by 45.4%, 24.4%, 37.2%, 44.9%, and 25.2% in the HK-2 (*p* < 0.01). Furthermore, compared to the cisplatin non-treated Veh group, the cisplatin-treated Veh group showed significant increases by 3.7 and 3.1 folds in the expression of p38-α and JNK2, respectively, in the NRK, and by 3.4 and 3.7 folds in the HK-2 models (*p* < 0.01). There were significant differences among the groups in the expression of p38-α in NRK (F = 116.0, *p* < 0.01) and HK-2 (F = 55.1, *p* < 0.01), as well as JNK2 in NRK (F = 77.7, *p* < 0.01) and the HK-2 (F = 92.9, *p* < 0.01). The expression of p38-α decreased by 15.1%, 24.5%, 19.7%, 28.1%, and 25.8% in the IJ, Uro, US, YJ, and AA groups, and by 16.2%, 19.4%, 21.7%, 27.2%, and 22.2%, respectively in NRK and HK-2 (*p* < 0.01). The JNK2 decreased by 24.2%, 31.4%, 26.1%, 38.1%, and 37.9% in the IJ, Uro, US, YJ, and AA groups, respectively, in the NRK, and by 18.1%, 31.4%, 36.2%, 24.8%, and 30.1% in the HK-2 (*p* < 0.01).

## 3. Discussion

Similar to other studies [21,22,23], we found that cisplatin showed IC_50_ values at 20 μM in NRK and 16 μM in HK-2, and ascorbic acid was the most efficient at 35 μg/mL (250 μM) in both cisplatin-induced cell injury models. The cell models showed increased ROS production and reduced activities of antioxidant enzymes, SOD and catalase, due to responses to the oxidative stress [24]. There have been accumulated pieces of evidence that cisplatin-induced nephrotoxicity involves oxidative damage to renal tubular cells and tissues [6,25]. Cisplatin reacts with an endogenous antioxidant, glutathione, and produces reactive electrophiles, which deteriorate mitochondrial function, and disrupt the electron transport chain, leading to increased ROS production [5,25]. The overproduced ROS reacts with lipids, proteins, and nucleic acids, causing oxidative stress and damage [26]. Here, the polyherbs showed significant free radical scavenging activities, and the treatments prevented cisplatin-induced cell death in both NRK and HK-2 models, especially at 1.2 mg/mL. Furthermore, the polyherbs reduced the ROS levels, which might increase the activities of SOD and catalase probably by the reduced consumption to the oxidative stress. These results were similar to those of the treatment of ascorbic acid used as a strong antioxidant. It is likely that the antioxidant properties of the polyherbs might contribute to reduce the oxidative stress and conserve antioxidant enzymes, resulting in protective effects against cisplatin-induced cell injuries. Many animal studies have shown nephroprotective effects via antioxidant activity [6,27], suggesting that antioxidant polyherbs can have therapeutic potential in AKI.

Our previous study has shown the nephroprotective effects of four other polyherbs including Bojungikki-tang, Palmijihwang-tang, Oryeong-san, and Wiryeong-tang [23]. The polyherbs are main traditional medicines for treating renal diseases; however, the antioxidant effects were little in treatments with Oryeong-san or lower in treatments with others. Given that the pathogenesis of AKI involves tubular oxidative stress, we selected the current polyherbs showing stronger antioxidant properties than those of Bojungikki-tang and Wiryeong-tang. However, because all the polyherbs used previously and currently inhibited cisplatin-induced apoptosis and cell death, the relevant pathway was further examined. Increased ROS is known to activate the transcription factor, NF-κB, which plays a key role in inflammatory progress [28]. Activation of NF-κB increases the levels of proinflammatory cytokines (i.e., tumor necrosis factor-α and interleukin-6) and pro-apoptotic proteins (Bcl-2 family), and mediates cell survival and differentiation [29]. The MAPK signaling pathway comprising p38, ERK, and JNK is an upstream component of NF-κB [30]. Furthermore, ROS also activates JNK in proximal tubular epithelial cells, and oxidative stress with activated JNK accelerates MAPK signaling, which induces renal cell apoptosis [31,32,33]. Indeed, activation of MAPK proteins has been associated with renal cell apoptosis and inflammation, leading to renal dysfunction [30,34]. In this context, the MAPK pathway has a significant correlation with the regulation of oxidative stress, apoptosis, and inflammation, and agents inhibiting oxidative stress and the MAPK pathway can have therapeutic potential for cisplatin-induced AKI. Here, a cisplatin-induced cell model showed increased expression of activated caspase-3 and apoptotic cell death as a result of the mitochondrial apoptotic cascade [35]. However, treatment with the polyherbs resulted in antiapoptotic effects, accompanied by inhibition of activation of MAPK proteins (p-38 and JNK). This indicates that the nephroprotective effects of the polyherbs involve the inhibition of oxidative stress and MAPK signaling.

In traditional Korean medicine, IJ is prescribed for treating inflammation-related diseases, especially in the liver, and Uro, US, and YJ, are used for treating kidney deficiency [20]. YJ, consisting of six herbs, is the most common polyherb used for treating renal diseases, and many reports have shown its preventive effects on renal hypertension and ischemic acute renal failure [36,37], along with antioxidant and anti-tumor effects [36,38,39]. US consists of 10 herbs, including one in YJ. The *Alismatis rhizome* and *Dioscoreae rhizome* contained in the YJ and US have been shown to improve blood flow [40] and to exert anti-inflammatory effects [41,42,43]. In addition, the main herb (*Artemisia capillaris*) of IJ and its compounds (dapillarisin) have shown strong antioxidant and anti-inflammatory effects [44,45]; Yondansagan-tang, which contains components similar to those of Uro, is used to treat hepatorenal syndrome with hepatic inflammation and dysuresia [46]. In addition, the single herbs of *Rhubarb* and *Gardenia* fruit included in IJ and Uro; *Cinnamon* in Uro and US; *Rehmannia* Root in Uro, US, and YJ; *Moutan* root bark in US and YJ; and *Cornus* fruit in YJ have been reported to exert nephroprotective effects in animal models [47,48]. However, because of a lack of mechanism studies, it is difficult to speculate which herbs or combinations can have nephroprotective effects. The present study is the first to report the nephroprotective effects of IJ, Uro, US, and YJ, and their relevant mechanisms. Because traditional polyherbal formulae are used in various herbal combinations based on accumulated clinical experiences, they may have synergic effects via multi-targeting. Furthermore, YJ possessing antitumor effects can be used in combination with chemotherapeutic agents for treating renal cancer-related AKI. Future studies are, thus, needed to clarify the exact mechanisms by which polyherbal combinations enhance their beneficial effects.

Traditional polyherbs have advantages in that they can be applied to clinical patients for a long time with few side effects. Here, polyherb-related cell death was not observed in the NRK and HK-2 cells; rather, cell proliferation was observed in HK-2 cells treated with US or YJ. As mentioned above, US and YJ share six herbs, and their specific components could contribute to their proliferative effects on human tubular cells; however, these components are unclear. Here, we demonstrated that antioxidant polyherbs might exert nephroprotective effects by regulating MAPK signaling. The polyherbs used were approved by the Korea Food and Drug Administration (FDA); however, their use has generally ceased in hospitalized patients with AKI. These results support scientific evidence for the effectiveness of polyherbs and their relevant mechanisms, which provide useful information for their clinical application in AKI.

## 4. Materials and Methods

### 4.1. Preparation of Traditional Polyherbs

The four polyherbs used here have been approved by the Korea FDA as general pharmaceuticals, and are commercially available. These were IJ (Panparu^TM^, Hanpoong Pharmaceutical Co. Ltd., Daejeon, Korea), Uro (Urofen^TM^, Hanpoong Pharmaceutical Co. Ltd.), US (Bosinji^TM^, Jeil Pharmaceutical Co. Ltd., Seoul, Korea), and YJ (Yunbohwan^TM^, Kyoungbang Pharmacy, Incheon, Korea). Their individual ingredients are listed in Table 1. The polyherbs were dissolved in absolute dimethyl sulfoxide (DMSO; Sigma-Aldrich, St. Louis, MO, USA), and then diluted with the cell culture medium at a final concentration of 2.4 mg/mL with 0.5% DMSO as a vehicle. They were filtered through a pore size of 0.22 μm, and stored at 4 °C in the dark until use. The effects of the polyherbs were compared with those of AA (Sigma-Aldrich) as a positive control.

### 4.2. Free Radical Scavenging Activity

Antioxidant activities of polyherbs were assessed using the DPPH (Sigma-Aldrich) assay. Briefly, the polyherbal solution was incubated with 0.4 mM DPPH solution in methanol at a final concentration of 1 mg/mL for 30 min in the dark. Distilled water containing 0.5% DMSO was used as the vehicle control. Absorbance was measured at 517 nm using an automated microplate reader (BIO-TEK, Winooski, VT, USA) and antioxidant activity was calculated using the following formula:$$\mathrm{Inhibition}\;(\%)\;=\;1-\frac{\left(\mathrm{Absorbance}\;\mathrm{of}\;\mathrm{control}\;-\;\mathrm{Absorbance}\;\mathrm{of}\;\mathrm{polyherb}\right)}{\mathrm{Absorbance}\;\mathrm{of}\;\mathrm{control}}\;\times 100$$

### 4.3. Cell Culture

Kidney proximal tubular epithelial cell lines, rat NRK-52E (NRK) and human HK-2, were obtained from the American Type Culture Collection (URL www.atcc.org). NRK and HK-2 were cultured in Dulbecco’s modified medium–high glucose (Hyclone, Logan, UT, USA) and Roswell Park Memorial Institute 1640 (Gibco, Grand Island, NY, USA), respectively. The media were supplemented with 100 U/mL penicillin/streptomycin and 10% fetal bovine serum (FBS; Gibco). The cells were maintained at 37 °C in a humidifying incubator with 5% CO_2_.

### 4.4. Cisplatin-Induced Acute Kidney Cell Model and Treatments

When cells were grown to 80–90% confluence, they were seeded in 96-well (1 × 10^4^ cells/well) and 6-well (1 × 10^6^ cells/well) plates. The AKI cell model was induced by cisplatin (Sigma-Aldrich) at 20 μM in NRK and at 16 μM in HK-2, as the IC_50_ under the serum-free conditions. The cells were then treated with the polyherbs for three days, and the results were compared with those of the negative control treated with vehicle alone (Veh).

### 4.5. Cell Viability Assay

Cell viability was assessed using a MTT (TCI Chemicals, Tokyo, Japan) assay. MTT solution at 0.5 mg/mL in distilled water was added to the treated cells and incubated for 1 h at 37 °C. The cells were then lysed in DMSO, and the absorbance was measured at 570 nm using a microplate reader (BIO-TEK). Viability was represented as a percentage of the Veh group.

### 4.6. Assessment of ROS Levels

Cells (1 × 10^4^ cells/well in a black 96-well flat-bottom plate) or cell suspensions (5 × 10^4^ cells/well in a black 96-well V-bottom plate) were incubated with 5 μM H2DCFDA (Invitrogen, Waltham, MA, USA) at 37 °C for 15 min. The stained cells were observed using an inverted fluorescence microscope, and the fluorescence intensities of the cell suspensions were measured at Ex-495nm and Em-520nm using a microplate reader (BIO-TEK).

### 4.7. Assessment of Activities of Antioxidant Enzymes

Activities of antioxidant enzymes, SOD and catalase, were measured using commercial enzyme-linked immunosorbent assay kits (#706002 for SOD and #707002 for catalase, Cayman, Ann Arbor, MI, USA), according to the manufacturer’s instructions. For SOD, cells were sonicated in 20 mM HEPES buffer (pH 7.2) containing 1 mM ethylene glycol tetraacetic acid, 210 mM mannitol, and 70 mM sucrose. The cell lysates were centrifuged at 1500× *g* for 5 min at 4 °C, and the supernatants were reacted with tetrazolium salt. For catalase, cells were sonicated in 50 mM potassium phosphate buffer (pH 7.0) containing 1 mM ethylenediaminetetraacetic acid. The lysates were centrifuged at 10,000× *g* for 15 min at 4 °C, and the supernatants were reacted with formaldehyde. The absorbance of the reactions was measured at 450 and 540 nm for SOD and catalase, respectively, under the standard curves using a microplate reader (BIO-TEK).

### 4.8. Flow Cytometric Analysis

Apoptotic changes and cell proliferation were measured using flow cytometry, as described previously [23]. Briefly, for apoptosis, cell samples were incubated with a rabbit anti-annexin V–FITC (1:1000, #14085, Abcam, Cambridge, UK) for 30 min, followed by propidium iodide at 50 μg/mL (Life Technologies, Carlsbad, CA, USA) for 10 min. For determining cell proliferation, cells were fixed in 4% formaldehyde solution, and cell membranes were permeated with saponin at 1 mg/mL (TCI chemicals). The cells were incubated with a rabbit anti-Ki-67 antibody (1:100, #15580, Abcam), and then with an Alexa 488-conjugated goat anti-rabbit IgG antibody (1:1000, #11008, Life Technologies) for 30 min each. All steps were performed on ice, and cells were washed three times with phosphate-buffered saline containing 2% FBS between each step. The cells omitting the primary antibodies were used as negative controls. The immunopositive cells were analyzed on a BD Accuri C6-Plus flow cytometer (BD Bioscience, San Jose, CA, USA).

### 4.9. Immunoblotting

Cells were centrifuged at 10,000× *g* for 10 min at 4 °C, and the cell pellet was lysed in RIPA buffer (Rock Land, Pottstown, PA, USA) with 1 mg/mL protease inhibitor (Leupeptin^TM^, Roche, Mannheim, Germany) for 30 min. After measuring the amount of total protein using the BCA assay (Thermo-Fisher, Rockford, IL, USA), the lysates were mixed with sodium dodecylsulfate (SDS)-gel loading buffer (Biorad, Hercules, CA, USA), and boiled for 10 min. Equal amounts of samples were electrophoresed on 10% SDS-polyacrylamide gel electrophoresis gels, and transferred onto nitrocellulose membranes using semi-dry blot transfer (Bio-Rad). The membrane was blocked with 5% skim milk in tris-buffered saline with 0.1% tween (TBST), and then incubated overnight at 4 °C with the following primary antibodies: mouse anti-cleaved caspase-3 (1:100, #9668, Cell Signaling, Danvers, MA, USA), mouse anti-p38-α (1:500, #8691, R&D systems, Minneapolis, MN, USA), rabbit anti-JNK2 (1:1000, #178953, Abcam), and mouse anti-β-actin antibodies (1:2000, Abcam). The next day, the cells were incubated with anti-mouse and anti-rabbit horseradish peroxidase-conjugated secondary antibodies (1:1000, #1706515 and #1706516, respectively, Biorad) for 1 h. The membrane was washed with TBST five times for 30 min after each incubation. The expression was visualized using WESTARηC2.0 (Cyanagen, Bologna, Italy), and analyzed using a ChemiDoc instrument (Bio-Rad). The expression levels were normalized to those of β-actin.

### 4.10. Statistical Analysis

Data are expressed as the means ± standard deviation in each experiment performed independently at least three times. The homogeneity of variance was examined by the Levene test. As it was not significant, multi-comparison ANOVA was examined, followed by Tukey post-hoc tests. Dose-dependent effects of polyherbs on the cell viabilities in the AKI model were examined by two-way ANOVA with main factors for the group and the dose, and the others were examined by one-way ANOVA. Multi-comparison was described to be significant in the treatment group compared to the Veh group. A *p*-value of less than 0.05 was considered statistically significant.

## Figures and Tables

**Figure 1 molecules-25-05641-f001:**
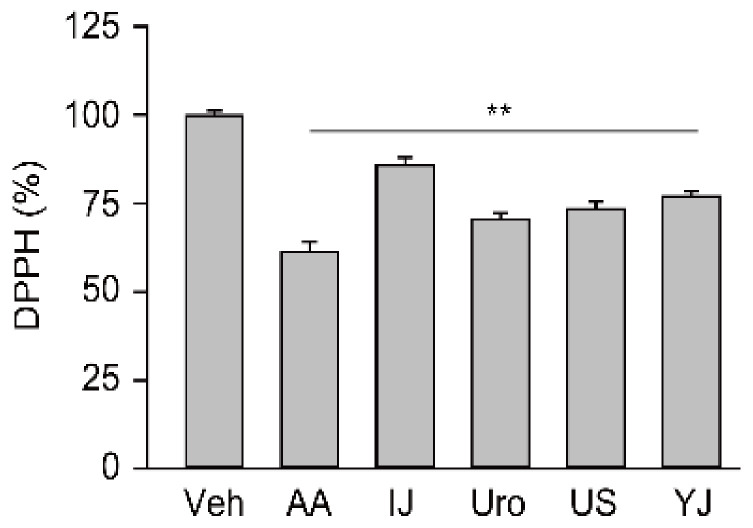
Free radical scavenging activity of polyherbs. Antioxidant activity of four polyherbs, IJ, Uro, US, and YJ, was assessed using the 1,1-diphenyl-2-picrylhydrazyl (DPPH) assay. Ascorbic acid (AA) was used as a positive control. The results were compared with the vehicle control (Veh), and were expressed as a percentage of the control. Values are means ± standard deviation (SD) from three independent experiments. **: *p* < 0.01 versus the Veh group.

**Figure 2 molecules-25-05641-f002:**
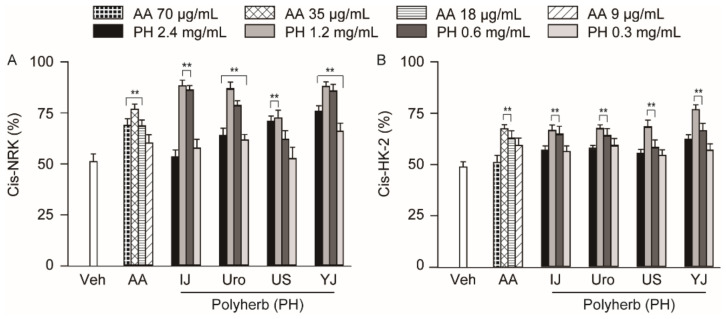
Protective effects on cisplatin-induced cell injuries. Renal cell injury was induced by cisplatin in normal rat kidney (Cis-NRK, (**A**)) and human kidney-2 (Cis-HK-2, (**B**)) cells. The cells were treated with AA, Injinhotang (IJ), UrofenTM (Uro), Ucha-Shinki-Hwan (US), or Yukmijihwang-tang (YJ) at the indicated doses for 3 days. The results are expressed as a percentage of the cell viability in the cisplatin non-treated normal control. Values are means ± SD from three independent experiments. **: *p <* 0.01 versus the vehicle group (Veh).

**Figure 3 molecules-25-05641-f003:**
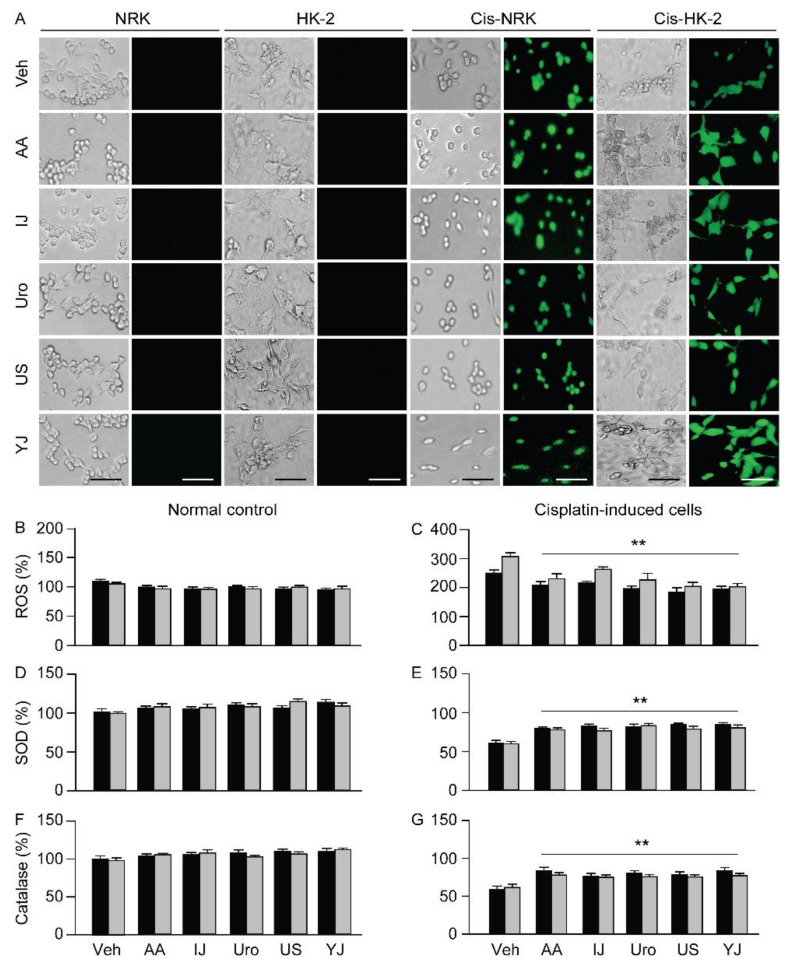
Effects on reactive oxygen species production and antioxidant activities. The cisplatin-induced NRK and HK-2 cells (Cis-NRK and Cis-HK-2, respectively) and the corresponding normal controls (NRK and HK-2) were treated with AA, IJ, Uro, US, or YJ for 3 days. Representative images of cells stained with dichlorofluorescein diacetate for reactive oxygen species (ROS) are shown in (**A**). Scale bars indicate 50 μm. Then, the levels of reactive oxygen species (ROS, (**A**–**C**)) and activities of the superoxide dismutase (SOD, (**D**,**E**)) and catalase (**F**,**G**) were assessed. Graphs show the results as percentages of the cisplatin-nontreated control. Values are expressed as means ± SD from three independent experiments. **: *p* < 0.01 versus the cisplatin-treated Veh group.

**Figure 4 molecules-25-05641-f004:**
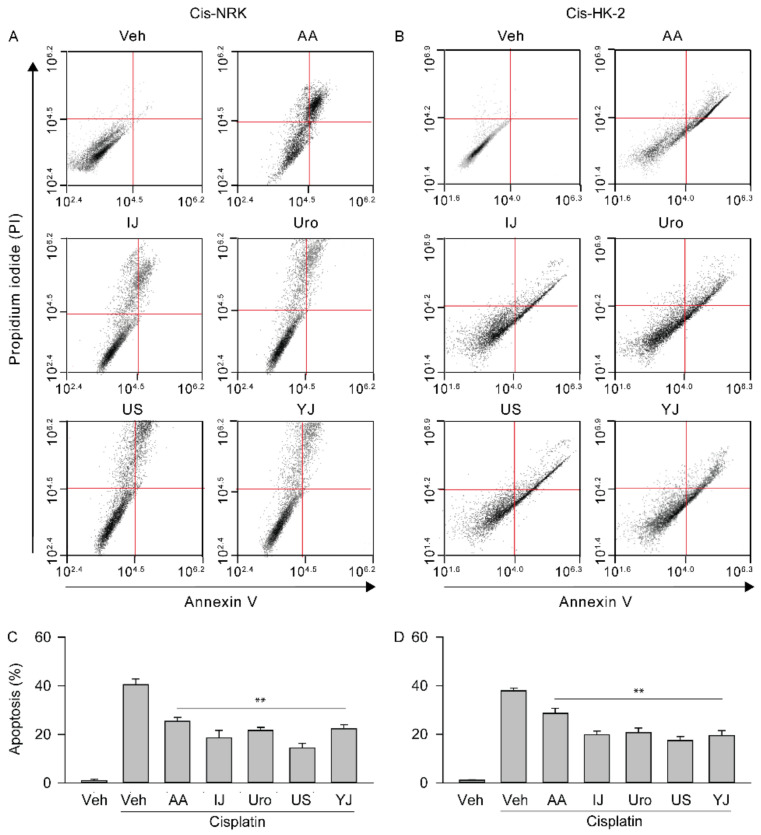
Effects on apoptotic changes. The cisplatin-induced NRK (Cis-NRK, (**A**)) and HK-2 (Cis-HK-2, (**B**)) cells were treated with AA, IJ, Uro, US, or YJ, and apoptosis was measured by the annexin V- and PI-double positive cells by flow cytometry. The results are expressed as percentages of the cisplatin non-treated normal control (**C**,**D**). Values are means ± SD from three independent experiments. **: *p* < 0.01 versus the cisplatin-treated Veh group.

**Figure 5 molecules-25-05641-f005:**
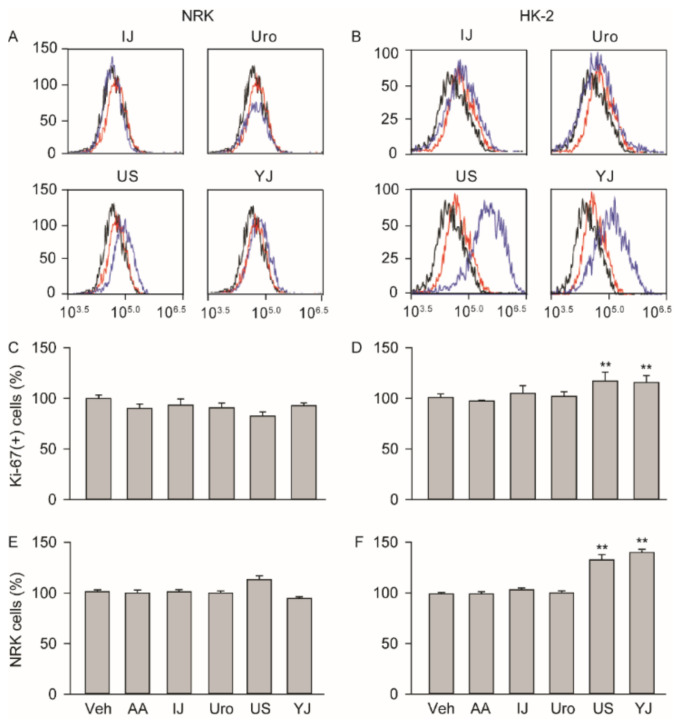
Effects on cell proliferation. Cells were treated with AA, IJ, Uro, US, or YJ under serum-free conditions for 3 days, and Ki-67-positive proliferative cells were measured in the NRK (**A**) and HK-2 (**B**) using flow cytometry. Blue and red lines indicate the polyherb treatment groups and vehicle control (Veh), respectively. Black indicates the experimental control omitting the primary antibody. Results are expressed as a percentage of the Veh group (**C**,**D**). In addition, the cell growth was assessed and expressed as a percentage of the Veh group (**E**,**F**). Values are means ± SD from five independent experiments. **: *p* < 0.01 versus the Veh group.

**Figure 6 molecules-25-05641-f006:**
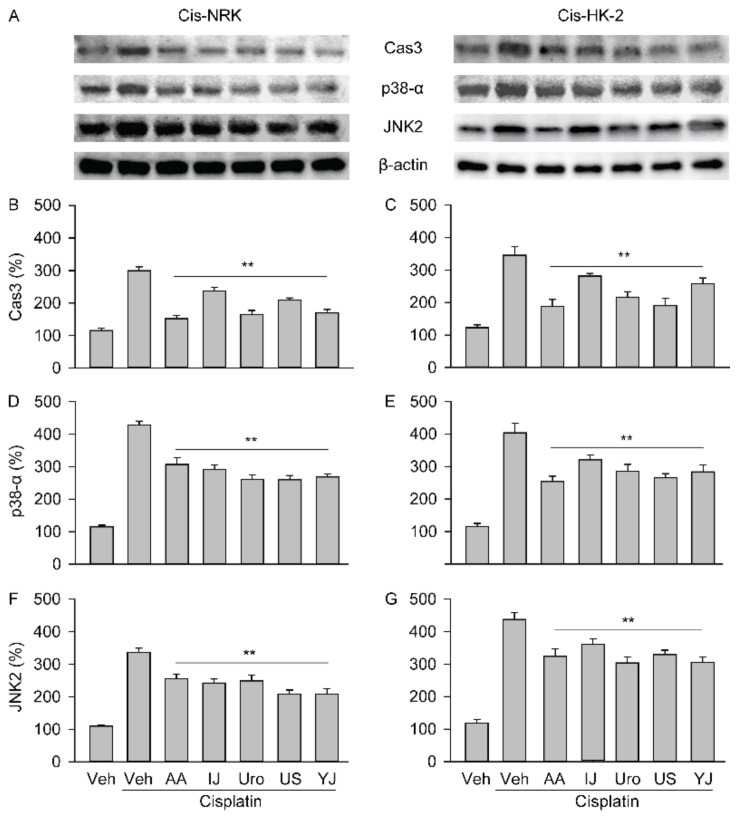
Effects on the expression of cleaved caspase-3 and activated mitogen-activated protein kinase (MAPK)-related proteins. The cisplatin-induced cells were treated with AA, IJ, Uro, US, or YJ for 3 days, and the expression of cleaved caspase 3 (Cas3) and phosphorylated MAPK proteins, p-38α and c-Jun N-terminal kinase (JNK2), was assessed using western-blotting (**A**). The expression was normalized to the levels of β-actin. The results are expressed as percentages of the cisplatin non-treated control (**B**–**G**). Values are means ± SD from three independent experiments. **: *p* < 0.01 versus the cisplatin-treated Veh group.

**Table 1 molecules-25-05641-t001:** Individual herbs composing the polyherbs used in this study.

	Ingredients
IJ	Artemisiae Capillaris Herba 2 g, Gardenia Fruit 1 g, Rhubarb 0.67 g
Uro	Akebiae Caulis 416.7 mg, Alisma Rhizome 250 mg, Angelica Gigas Root 416.7 mg, Cinnamon Bark 16.7 mg, Ephedra Herb 16.7 mg, Forsythia Fruit 16.7 mg, Gardenia Fruit 125 mg, Gentian Root 125 mg, Glycyrrhiza 125 mg, Plantago Seed 250 mg, Rhubarb 16.7 mg, Scutellaria Root 250 mg, Raw Ginger 16.7 mg, Rehmannia Root 416.7 mg
US	Achyranthes Root 3.0 mg, Alisma Rhizome 3.0 mg, Cinnamon Bark 1 g, Cornus Fruit 3.0 mg, Dioscorea Rhizome 3.0 mg, Hoelen 3 g, Moutan Root Bark 3.0 mg, Psyllium Husk 3 g, Pulvis Aconiti Tuberis Purificatum 1.0 mg, Rehmannia Root 5.0 mg
YJ	Alisma Rhizome 240 mg, Cornus Fruit 320 mg, Dioscorea Rhizome 320 mg, Hoelen 240 mg, Moutan Root Bark 240 mg, Steamed Rehmannia Root 640 mg

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
