# Peer review of "Protective Effects of Traditional Polyherbs on Cisplatin-Induced Acute Kidney Injury Cell Model by Inhibiting Oxidative Stress and MAPK Signaling Pathway"

_molecules, 2020, doi:10.3390/molecules25235641_

Round 1
Reviewer 1 Report
Major comments:
- Although the composition of the herbal mixtures studied are given, it is not known how they were prepared so these studies are hardly reproducible in other laboratories. The English literature on these herbal mixtures is very scarce, e.g. there is just one hit in Scopus for "Injinhotang" and "Ucha-Shinki-Hwan" keywords and no hit for "„Yondansagan-tang” keyword.
- In most experiments the beneficial effects of the herbal mixtures are similar or slightly better than those of ascorbic acid, and in the case of the radical scavenging even worse.
- What do differently colored bars indicated as AA in panels A and B of Figure 2 mean? According to the legend they should mean a mixture of AA and some polyherb, which is contradictory with the meaning of other bars to the right.
- Description of the preparation of the herbal samples is too laconic. It is not known if the commercial mixtures are completely soluble (that would mean they are herbal extracts, and not dried herbs which are not expected to dissolve) or not. If not, how long they were extracted with DMSO?
Minor comments:
- Page 4, sentence „…the ROS increased by 2.3 and 2.7 folds…” – second from the left bars in panels A and B of Figure 3 indicate different values (higher).
- The langague of the paper needs extensive correction. Even in the title there is a strange expression "effect ... on ... cell model".
- While using the term „polyherbal” is well justified when meaning „composed of many herbs, the word „polyherb” sounds strange and it is scarcely used in the scientific literature. "Herbal mixture" may be a better choice.
Author Response
Please see the attachment
Major comments
Point 1. Although the composition of the herbal mixtures studied are given, it is not known how they were prepared so these studies are hardly reproducible in other laboratories. The English literature on these herbal mixtures is very scarce, e.g. there is just one hit in Scopus for "Injinhotang" and "Ucha-Shinki-Hwan" keywords and no hit for "„Yondansagan-tang” keyword.
Response 1: We purchased the commercial products for the polyherbs and dissolved them. We described the information for the product name and the company on lines 280-284 in page 9 (REVISED Version). The commercial products of polyherbs are approved by the Korea FDA based on the clinical experiences accumulated in Korean traditional medicine for a long time rather than the scientific evidences. So, there have been few reports in renal disease, and there are difficulties to find the relevant manuscript. You can find papers related to the Ucha-Shinki-Hwan and Yondansagan-tang, Chinese or Japanese name: Ucha-Shinki-Hwan using a Japanese name (Gochajinkigan) and Yondansagan-tang using a Chinese or Japanese name (Longdanxiegan-tang in Chinese; Ryutanshakan-to in Japanese), as below. However, a few reports examined the effects of Injinhotang, and they are written in Korean (Yoon and Kim, 2009, ‘The Effect of the Injinhotang Extract on Hepatocarcinogenesis in Rats’; Kim et al., 1998, ‘A Study on the Degraded Effect of Decoctedc Injinhotang over a Period’).
Point 2. In most experiments the beneficial effects of the herbal mixtures are similar or slightly better than those of ascorbic acid, and in the case of the radical scavenging even worse.
Response 2: There were no significant differences in all of the results in the treatments with polyherbs compared with those in the AA group. Although the beneficial effects of polyherbs were mostly similar with those of ascorbic acid, the polyherbs have been used for treating renal diseases in traditional Korean medicine. So, we expect that the polyherbs contain nephron-protective components involved in the therapeutic mechanisms such as the anti-apoptotic and anti-inflammatory properties as well as the antioxidant effects. The next in-vivo study is on progress to reveal the preventive effects on AKI.
Point 3. What do differently colored bars indicated as AA in panels A and B of Figure 2 mean? According to the legend they should mean a mixture of AA and some polyherb, which is contradictory with the meaning of other bars to the right.
Response 3: The different colours of the bars indicated the treated concentrations of AA or each polyherb. However, we agree that it is confusing, so we fixed the figure and figure legend.
Point 4. Description of the preparation of the herbal samples is too laconic. It is not known if the commercial mixtures are completely soluble (that would mean they are herbal extracts, and not dried herbs which are not expected to dissolve) or not. If not, how long they were extracted with DMSO?
Response 4: The polyherbs are commercially available as a powder-form of general pharceuticals, and they could be dissolved up to 500 mg in absolute DMSO. Then, we diluted the polyherb solutions with cell culture medium. We added more information on lines 284-286 in page 9 (REVISED Version).
Minor comments
Point 1. Page 4, sentence „…the ROS increased by 2.3 and 2.7 folds…” – second from the left bars in panels A and B of Figure 3 indicate different values (higher).
Response 1: We are sorry for the mistakes. We changed ‘2.3 and 2.7 folds’ to ‘2.5 and 3.1 folds’ on line 127 in page 4 (REVISED Version).
Point 2. The langague of the paper needs extensive correction. Even in the title there is a strange expression "effect ... on ... cell model".
Response 2: We improved our manuscript by English editing with a professional English editing service, Editage (www.editage.com). The changes are shown as blue.
Point 3. While using the term „polyherbal” is well justified when meaning „composed of many herbs, the word „polyherb” sounds strange and it is scarcely used in the scientific literature. "Herbal mixture" may be a better choice.
Response 3: As mentioned above, we improved our manuscript with scientific expressions. However, we have used the word ‘polyherbs’ in our papers, and we could find the expression in other papers. So, we would like to use the expression. However, we thank you for a kind advice.

Reviewer 2 Report
The authors have investigated potential protective effects of traditional polyherbs on two cisplatin-induced acute kidney injury cell models with an emphasis on oxidative stress and MAPK signaling pathway effects.
Major comments:
- The authors should clearly describe in the introduction, and discuss in the Discussion section, how their current study differs from their earlier publication “Effects of Traditional Herbal Formulas on Cisplatin-Induced Nephrotoxicity in Renal Epithelial Cells via Antioxidant and Antiapoptotic Properties”; doi: 10.1155/2020/5807484. The authors need to reduce redundancies and explain the differences between this publication and their current manuscript. They should clearly point out and explain how their current manuscript adds to the content that already has been published.
- With respect to the results shown in line 138, figure 3. The same experiment should be performed without cisplatin treatment to investigate a possible mechanism: is the antioxidant enzyme activity increased by polyherb treatment, e.g. by upregulation of enzyme expression, or is the antioxidant enzyme activity, by polyherb treatment, protected against oxidative damage?
- How do the concentrations of the polyherb formulations used in the current study correspond to concentrations achieved by traditional routs of application for such polyherb formulations?
Minor comments:
- Line 95, figure 1: Is “Con” in this figure the same as “Veh” in the rest of the manuscript?
- Line 111/112: The concentrations should be given in a consistent way for all substances.
- Line 128: The meaning of this sentence is unclear.
Author Response
Please see the attachment
Major comments
Point 1. The authors should clearly describe in the introduction, and discuss in the Discussion section, how their current study differs from their earlier publication “Effects of Traditional Herbal Formulas on Cisplatin-Induced Nephrotoxicity in Renal Epithelial Cells via Antioxidant and Antiapoptotic Properties”; doi: 10.1155/2020/5807484. The authors need to reduce redundancies and explain the differences between this publication and their current manuscript. They should clearly point out and explain how their current manuscript adds to the content that already has been published.
Response 1: Yes, you are right. The previous study is similar with current one because the objectives and methods were similar. We described why these polyherb candidates were selected for nephroprotective properties on lines 79-86 in page 2 and 225-232 in page 7 and 8. The polyherbs used in the previous study are mostly well-known drugs for renal diseases in traditional Korean medicine, so we first investigated that the polyherbs have the beneficial effects. In addition, we found the other traditional polyherbs showing stronger antioxidant activities than the previous one. So, we used them for nephroprotective effects in this cell model for comparing the beneficial effects later, even though the experiments were not carried out simultaneously. Here, we found similar nephroprotective effects of the antioxidant polyherbs, and further examined the pathways. Indeed, we are examining what is different between both polyherb matching in previous and current studies. For example, we are examining the signaling pathway in the treatments of OR (previously showing proliferating effects) and YJ (currently showing strong antioxidant effects) by in vivo study and next generation sequencing (NGS) data analysis. All of these herbal formulas showed significant antiapoptotic activities. The relevant signaling pathways might result in the protective effects on the renal cell injury. We consider that the efficacy can involve STAT3, Nrf signaling or NF-κB signaling pathways. The NGS analysis is a work in progress in the renal epithelial cells treated with OR or YJ. The results showed that OR treatments upregulated genes involving mitotic cell cycle and process and nuclear/cell division. Now, we are assessing the expressions of each top 10 genes showing up- or down-regulation. It is interesting because OR can be used as a regenerative medicine in AKI patients if it stimulates the signaling pathways involving the renal cell proliferation. Our final goal is to find the best herbal combinations that can be applied to the clinical trials in AKI.
Point 2. With respect to the results shown in line 138, figure 3. The same experiment should be performed without cisplatin treatment to investigate a possible mechanism: is the antioxidant enzyme activity increased by polyherb treatment, e.g. by upregulation of enzyme expression, or is the antioxidant enzyme activity, by polyherb treatment, protected against oxidative damage?
Response 2: We examined effects of the polyherbs on the activities of antioxidant enzymes, however, there were no differences in the treatments compared to the control group. It means that the free radical scavenging effect of the polyherbs may result in the increased levels of SOD and catalase in the cisplatin-induced model. We changed Fig. 3 and added the relevant information on lines 144-149 in page 4 and 5, on lines 122-125 in page 3, and on line 141-142 in page 4.
Point 3. How do the concentrations of the polyherb formulations used in the current study correspond to concentrations achieved by traditional routs of application for such polyherb formulations?
Response 3: It is ideal to match the concentration of polyherbs corresponding to the clinical applications. The polyherbs are usually applied via an oral route, however, this polyherbs were co-treated in the cell medium, so we examined the greater effects of the polyherbs on cell viabilities in the cisplatin-induced cell toxicity. The nephroprotective ranges of the concentrations were different, however, the cell viabilities were greater at 1.2 mg/ml in all of the polyherbs than the other concentrations. So we treated the polyherbs at 1.2 mg/ml.
Minor comments
Point 1. Line 95, figure 1: Is “Con” in this figure the same as “Veh” in the rest of the manuscript?
Response 1: Yes, it is same. We changed ‘Con’ to ‘Veh’ on lines 97 and 99 in page 3.
Point 2. Line 111/112: The concentrations should be given in a consistent way for all substances.
Response 2: The cell protective effects against the cisplatin-induced cell toxicity were significantly different in the PH groups depending on the doses. Although the cell viabilities were significantly different depending on the dose ranges, they were greater at a dose of 1.2 mg/ml in all the polyherbs.
Point 3. Line 128: The meaning of this sentence is unclear.
Response 3: In the statistical analyses, the post-hoc tests for multi-comparison can be analyzed only in cases of significant differences among the groups. In the one-way ANOVA, the level of ROS (shown in Fig. 3A) was significantly different among the groups, so we compared the levels in the treatment groups compared with those of the control group. For better understanding, we improved our manuscript by English editing with a professional English editing service, Editage (www.editage.com). The changes are shown as blue.

Reviewer 3 Report
induced acute kidney injury cell model by inhibiting oxidative stress and MAPK signaling pathway. Authors have established the role of polyherbs in ameliorating the cisplatin induced AKI. I have some major and minor concerns related to experimental design, Major concerns
- I strongly believe that authors should inject mice with polyherbs and show changes in morphology, improvement in renal functions. Most of the times invitro changes are hard to replicate in invivo system.
- It would be good have some images reflecting ROS generation (DCF, DHE, mitQ) in cells.
Minor revision
- Manuscript has lot of typos errors
- Authors should potentially consider revising manuscript in terms of English, specifically result (Line number 185,186) and discussion (199) section. Consider writing more scientifically.
Author Response
Major comments
Point 1. I strongly believe that authors should inject mice with polyherbs and show changes in morphology, improvement in renal functions. Most of the times invitro changes are hard to replicate in invivo system.
Response 1: Yes, we strongly agree with your opinions. Although there are many natural products and compounds reporting the therapeutic potentials in in-vitro studies, the results were inconsistent in in-vivo studies or clinical applications. So, we have screened the traditional polyherb formulas used for treating patients with renal diseases for a long time. The polyherbs are approved from the Korean FDA as general drugs based on the clinical experiences rather than the scientific evidences, and they were commercially available. However, the effectiveness is unclear. Previously, we screened the nephroprotective effects of traditional polyherbs in this cell model, and we are examining the therapeutic effects of BJ, OR and Hexa in cisplatin-induced mice model. Similarly, we also plan to examine the beneficial effects of the current polyherbs in the cisplatin-induced mice model. There are difficulties to examine lots of the traditional polyherbs in vivo simultaneously, so we are screening the best candidates for the future in-vivo and clinical studies.
Point 2. It would be good have some images reflecting ROS generation (DCF, DHE, mitQ) in cells.
Response 2: We added the images in Fig. 3A, and added the information on lines 122-125 in page 3 and on lines 144-149 in page 4 (REVISED Version).
Minor comments
Point 1. Manuscript has lot of typos errors
Response 1: We improved our manuscript by English editing with a professional English editing service, Editage (www.editage.com). The changes are shown as blue.
Point 2. Authors should potentially consider revising manuscript in terms of English, specifically result (Line number 185,186) and discussion (199) section. Consider writing more scientifically.
Response 2: We did our best to express scientifically in our manuscript, together with English editing.
Round 2
Reviewer 1 Report
I am satisfied with the answers of the authors and the corrections made.
Reviewer 2 Report
In Table 1 the ingredients and the polyherb name need a better matching to enable clarity. Right now it looks as if for example Gardenia fruit is included two times in IJ.